# OpenReview forum: "Preserving Ignorance Awareness in LLM Fine-Tuning"
_ICLR.cc/2026/Conference — Submitted to ICLR 2026_

### Official Review · Reviewer_ZnQP · 2025-10-27

**Soundness:** 3
**Presentation:** 3
**Contribution:** 3
**Rating:** 6
**Confidence:** 2

**Summary:**

This paper points out that fine-tuning can make LLMs lose their ability to say “I don’t know.”
It shows that normal fine-tuning shifts activations and blurs the line between what the model knows and doesn’t know.
To fix this, the authors propose SEAT, a sparse and entity-aware tuning method that keeps knowledge boundaries clear and better preserves uncertainty while keeping performance strong.

**Strengths:**

Novel problem formulation: The notion of Ignorance Awareness and its degradation through fine-tuning is both original and safety-relevant.

Principled theoretical grounding: Provides formal definitions, propositions, and Lipschitz-bounded theorems linking sparsity to activation stability.

Strong empirical validation: Convincing results across datasets and models (Llama3-8B, Qwen2.5-7B) with ablations clearly isolating each SEAT component.

**Weaknesses:**

The approach’s scalability and implementation practicality for multi-round continual learning are not evaluated.

The work focuses on knowledge acquisition scenarios; the broader effects on refusal calibration or harmful prompt alignment remain unexplored.

**Questions:**

Can SEAT be extended to refusal calibration or safety alignment tasks, where the goal is to preserve ethical uncertainty rather than factual ignorance, as mentioned in the main paper?

---

> ### Author Response · Authors · 2025-11-28
>
> We thank the reviewer for their thoughtful feedback and for recognizing that our work (1) **novel problem formulation**, (2) **principled theoretical grounding** including formal definitions, propositions, and Lipschitz-bounded theorems linking sparsity to activation stability, (3) **strong empirical validation**. We hope the following responses address your concerns.
>
> ---
>
> **[W1]** Multi-round CL scenarios would have limited impact on IA retention. Even with random masking, where the effective sparsity ratio may decrease over multiple rounds, the resulting drop in IA capability is minor (see Table 3). Moreover, additional techniques can be applied to maintain a fixed sparsity level while continuing to update the same trainable parameter subset for new tasks. In such settings, users may also consider integrating traditional CL methods to mitigate interference between successive tasks. We will include a discussion of these extensions in the next version.
>
> ---
>
> **[W2]** Our motivation for proposing SEAT as a robust fine-tuning method that preserves aligned ignorance awareness (IA) stems from prior work showing that traditional fine-tuning for knowledge acquisition is particularly prone to collapsing IA alignment, while having comparatively limited impact on harmful alignment degradation [1].
>
> Given that knowledge acquisition is a central and widespread fine-tuning objective across industry use cases (e.g., [2]), and that IA is critical in mission-critical applications (see line 36), there is a clear need for fine-tuning methods that can acquire new knowledge effectively while preserving IA.
>
> While the EP component in SEAT is specifically designed to retain IA, the sparse training component plays a role by anchoring broader alignment in the activation space, and would also help mitigate degradation in harmful prompt alignment. We agree that further investigation into this broader effect is a valuable direction and encourage future work to explore it more deeply.
>
> [1] LLM Unlearning via Neural Activation Redirection. NeurIPS 2025.
>
> [2] A Comparative Analysis of Instruction Fine-Tuning LLMs for Financial Text Classification
>
> ---
>
> **[Q1]** Please see our response to W2.

---

### Official Review · Reviewer_8Sdd · 2025-10-29

**Soundness:** 1
**Presentation:** 2
**Contribution:** 1
**Rating:** 2
**Confidence:** 4

**Summary:**

Fine-tuning an already aligned language model deteriorates its tendency to reply “I don’t know” to a question it does not know the answer to. This paper proposes SEAT, a method that updates a randomly masked subset of model parameters at each gradient step, and minimizes the KL divergence between the original and fine-tuned model on perturbed variants of instances with entity names replaced. Experiments on a GPT-4o-generated dataset of real-world events, and on two synthetic datasets (TOFU and PISTOL) show that SEAT can preserve more of the ability to acknowledge not knowing than conventional fine-tuning.

**Strengths:**

1. Language models should indeed acknowledge ignorance when applicable; this is an important problem to study.
2. Isolating an ‘ignorance’ direction in model activations might inform analyses of its behavior.

**Weaknesses:**

I will group my critiques into three sections.
1. **Setting and problem**.

    a. The proposed issue is a non-problem. One is typically worried about continual training of base (pretrained) and not instruction-tuned models, since we can always perform light alignment/RL/… at the end. If it is claimed that base models can acknowledge lack of knowledge, then one should work with base models (Llama-3.1-8B or Qwen-2.5-7B, say). The proposed issue only exists—as far as the paper shows—with instruction-tuned versions (Llama-3-8B-Instruct and Qwen-2.5-7B-Instruct).

    b. Therefore the connection with “continual learning” category, as the related work section suggests, is quite tenuous. Even if it were in this category, several references are missing—for example, the original definitions of catastrophic interference [1,2], or even the original replay methods [3].

2. **Method**

    a. I don’t think the theoretical results amount to a meaningful result. Equations 3-5 basically (1) assume that the loss is l-Lipschitz (ok) (2) bound the difference in activation in terms of the supremum of the gradient norm (an incredibly loose upper bound that may not even exist—it only exists here because they assume that the parameters cannot grow apart more than a fixed amount) (3) claim that this upper bound will be lower if less parameters are updated since fewer of them will now change. This result does not mean much—I might as well say that no difference will be observed if I do not update any parameters; the set of equations is meaningless unless optimized and solved under some constraint of performance preservation.

    b. It is a big assumption to assume that the data is neatly packaged in terms of (subject, relation, object) tuples and also that one can find other, reasonable, replacements object’ to form perturbations. Most natural data is not packaged in a neat format as this, and finding a suitable replacement may not always be feasible; and of course, there might be a massive computational cost to rewriting the data in this format based on, e.g., an LLM.
Therefore, the method proposed here is not practical.

3. **Experiments**

    a. No baselines are compared to. In fact, even early on, the paper mentions that LoRA is insufficient, so at least LoRA must be included as a baseline. Additional baselines might include continual learning methods like replay techniques.

    b. Other sparse training methods should also be compared to, including, e.g., model editing methods.

[1] Michael McCloskey and Neal J. Cohen. “Catastrophic interference in connectionist networks: The sequential learning problem”. Psychology of Learning and Motivation - Advances in Research and Theory, 24:109–165, 1989.

[2] R. Ratcliff. “Connectionist models of recognition memory: Constraints imposed by learning and forgetting functions”. Psychological Review, 97(2):285–308, 1990.

[3] Anthony Robins. “Catastrophic forgetting, rehearsal and pseudorehearsal”. Connection Science, 7(2):123–146, 1995.

**Questions:**

1. Did you try your experiments with base (non-instruct) language models?

2. How does LoRA perform on the three benchmarks?

---

> ### Author Response · Authors · 2025-11-28
>
> We thank the reviewer for their thoughtful feedback and for recognizing that our work (1) addresses a **meaningful problem**, and (2) offers **informed analysis**. We hope the following responses address your concerns.
>
> ---
>
> **[W1]** We appreciate the reviewer’s comments and the opportunity to clarify both the setting and motivation of our work.
>
> **(a)** Regarding the concern that the proposed issue is a “non-problem,” we respectfully disagree. We think the confusion likely stems from the meaning of “base model” used in our original submission. To clarify, we use “base model” to refer to the instruction-tuned model released by model developers, in which critical alignment capabilities (including IA) have already been instilled. We intentionally avoid referring to it as a pretrained model, as post-training alignment has already been applied to endow the model with these capabilities beyond the initial pretraining phase.
>
> Instruction-tuned models (e.g., Llama3-8B-Instruct) are commonly used as starting points for downstream fine-tuning in real-world applications (e.g., incorporating private knowledge or task-specific customization [1]). In these critical settings, users specifically choose these models for their aligned capabilities, making the retention of Instruction Awareness (IA) after fine-tuning essential.
>
> The practical challenge we address is that such fine-tuning can significantly degrade pre-existing aligned behavior - IA. As we show in our experiments, even modest supervised fine-tuning leads to a collapse of refusal behavior on unverifiable or post-cutoff knowledge, a regression that is safety-critical and thus warrants attention. This problem has not been systematically studied in prior work and, as such, we believe our work fills an important gap.
>
> While one might argue that re-alignment after fine-tuning could restore these capabilities, we argue this is impractical for several reasons: (1) **re-alignment are often brittle**, frequently failing to fully recover the model’s original alignment, introducing secondary forgetting of the fine-tuning data or harming downstream utility [2, 3, 4] (e.g., Light re-alignment methods such as R-tuning fail to fully recover IA, as shown in our experiments in Table 2, Section 6.2); (2) re-alignment **typically requires access to high-quality proprietary alignment data, which is unavailable to most practitioners**; (3) the **added complexity and computational cost** of re-alignment make it impractical in resource-constrained settings.
>
> Hence, our objective is to **preserve the original aligned capability** of instruction-tuned models during fine-tuning, an objective that is both practically important and distinct from existing work.
>
> **(b)** Regarding the connection to continual learning: we agree that our setting differs from classical CL scenarios where task boundaries are explicitly defined, and our problem setting **extends CL to the preservation of existing capability**.
>
> We thank the reviewer for pointing out early foundational works in CL. We have updated the related work section to include the suggested foundational works, and have clarified the conceptual alignment and distinctions between our setting and standard CL frameworks.
>
> [1] A Comparative Analysis of Instruction Fine-Tuning LLMs for Financial Text Classification
>
> [2] Learning and Forgetting Unsafe Examples in Large Language Models
>
> [3] Do As I Do (Safely): Mitigating Task-specific Fine-tuning Risks in Large Language Models
>
> [4] Safety-alignment Reserved Low-rank Adaptation

---

> > ### Author Response · Authors · 2025-11-28
> >
> > **[W2]**
> >
> > **(a) Theoretical result**. We appreciate that the reviewer agrees with Lipschitz property. We agree that the bound in original Theorem 2, which relates activation displacement to parameter movement, is a worst-case upper bound - as is typical in local stability analyses. The existence of the bound is ensured by restricting parameter movement within a norm-bounded region, which SEAT enforces via standard practical mechanisms: RMSNorm, weight decay, and moderate learning rates. This was discussed in Corollary 4 and the proof of original Theorem 3, and further clarified in the revised version.
> >
> > Regarding the comment that *“trivially freezing all parameters leads to zero change (but also no learning)”*, we respectfully disagree with its applicability in our context, as it contradicts the core objective of fine-tuning: the model must learn from the fine-tuning dataset while preserving key aligned behaviors.
> >
> > SEAT freezes a subset of parameters and allows the remaining ones to adapt. This balances the retention of pre-aligned capabilities with the acquisition of new knowledge - a principle supported by the lottery ticket hypothesis and prior sparsity-aware learning literature.
> >
> > Empirically, our FT scores remain near 1.0 across all settings (Table 2), indicating that SEAT does not impair learning on the target task.
> >
> > To further clarify the connection to IA preservation, we have added new analyses in Figure 3(a) and (b), along with accompanying discussion in the Ablation Study section. Specifically, we measure both the absolute activation drift and its projection along the IA direction for fine-tuning and unseen datasets. The results show that **sparsity does not impede learning**: **activations of fine-tuning data exhibit meaningful shifts**, confirming that the remaining trainable parameters are sufficient to adapt. In contrast, **change of activations for unseen data remain close to zero**, demonstrating strong retention of the model’s original IA capability.
> >
> > **(b) Practicality of assuming (s, r, o) triples.** To clarify, SEAT **does not require the input or output data to be explicitly formatted as (s, r, o) triples**. This phrasing was used to illustrate the most common and intuitive case, as many fine-tuning datasets (especially those derived from knowledge structured databases of industry practitioners) naturally reflect this structure. However, SEAT only requires that the prompt mentions a subject entity. This approach is broadly applicable, as any meaningful instruction (e.g., “Tell me about [subject]”) necessarily involves a subject entity, which is indispensable to the prompt’s intent.
> >
> > Additionally, SEAT does not impose any constraints on the format or type of the response. It works equally well whether the answer is a short factual string (as in PISTOL) or a long-form paragraph (as in TOFU). The perturbation mechanism operates at the prompt level, and only requires substituting a random subject span, without requiring a LLM rewrite.
> >
> > We thank the reviewer for raising this and have clarified in the revised Section 4.

---

> > > ### Author Response · Authors · 2025-11-28
> > >
> > > **[W3]** As suggested by the reviewer, we have added LoRA, replay techniques and other continual learning baselines in revised Table 2.
> > >
> > > The findings substantiate our claim: When applying LoRA, CLoRA, EWC or R-tuning, the model almost completely loses IA capability, with IDK rates on the unverifiable-knowledge dataset dropping from ~1.0 (base model) to less than 0.1 in most circumstances. Experience replay shows greater robustness in preserving IA, its performance remains significantly below SEAT, unstable, and highly dependent on the base model and fine-tuning dataset. This suggests that while traditional CL methods are effective for task-level knowledge retention, they lack robustness in preserving original IA alignment.
> > >
> > > In the context of model editing, base models are aligned to express ignorance in a dynamic and context-aware manner. This makes it infeasible to edit specific output tokens away from refusal. Moreover, the activations corresponding to these abstention behaviors cluster with other prompts where the model should remain abstain. As a result, adapting existing model editing techniques to operate in this latent space is also impractical.
> > >
> > > These results strengthen our core message: preserving alignment-level meta-cognitive behaviours requires a method fundamentally different from traditional CL strategies, and SEAT explicitly targets this under-explored yet practically crucial capability. We have updated the experimental section to better emphasize this distinction and added the new baselines for completeness and clarity.
> > >
> > > We hope this expanded comparison adequately addresses the reviewer’s concerns and demonstrates that SEAT is not merely competitive but uniquely effective for the targeted problem setting.
> > >
> > > ---
> > >
> > > **[Q1]** Our goal is to preserve ignorance awareness (IA) as an aligned capability already instilled (original submission line 45) in the base model (i.e., the instruct model). Therefore, pretrained models are not the focus of our experimental setting.
> > >
> > > **[Q2]** All benchmark metrics drop to near zero for LoRA, indicating a complete collapse of the model’s IA capability. This is further supported by the PCA analysis in Figure2(c), which shows no separation between seen and unseen data along the IA direction in the latent space of the LoRA fine-tuned model.

---

### Official Review · Reviewer_ytXg · 2025-10-30

**Soundness:** 3
**Presentation:** 3
**Contribution:** 2
**Rating:** 4
**Confidence:** 4

**Summary:**

This paper tackles the problem of preserving Ignorance Awareness (IA), a model’s ability to recognize when it does not know, during fine-tuning. The authors argue that standard fine-tuning disrupts epistemic boundaries, causing models to produce overconfident or hallucinatory answers. They propose Sparse Entity-aware Tuning (SEAT), combining sparse parameter updates and entity perturbation to maintain IA while retaining task performance.

**Strengths:**

1. The paper addresses an underexplored yet important aspect of model alignment (epistemic humility) and proposes a clear conceptual framework to formalize it.

2. The theoretical analysis provides a general justification for stability under sparse updates, formalizing how smaller parameter movements constrain activation drift.

3. Experiments demonstrate consistent improvements on IA-related metrics, showing practical relevance.

**Weaknesses:**

1. The framework implicitly assumes that large pretrained models already possess a “refusal-to-answer”. Demonstrating that the pretrained model indeed encodes this capability requires more direct evidence. Of course, the paper demonstrates that the pretrained model possesses a refusal capability, but only on synthetic data. It would strengthen the work if the authors could also show evidence that such capability exists when dealing with real-world entities or natural data.

2. The experimental setup assumes a difficult fine-tuning scenario where only task-specific data are available. While this setting is realistic, it also restricts generality.

3. The theoretical proofs are mathematically consistent but very general. They describe overall knowledge preservation through limited parameter movement, not IA-specific stability. The masking operation may preserve all representations, including those that should adapt, potentially impeding learning. It would help to explicitly connect the mathematical derivation and method to IA-specific features or scores.

4. The empirical section lacks strong baselines. Random masking, although theoretically justified (Corollary 1-2), contributes little beyond a regularization effect. More meaningful comparisons with alternative stability-preserving methods (e.g., LoRA variants, EWC, or alignment-oriented fine-tuning such as DPO or ORPO) are necessary to demonstrate the distinctiveness of SEAT.

**Questions:**

See Weekneses.

---

> ### Author Response · Authors · 2025-11-28
>
> We thank the reviewer for their thoughtful feedback and for recognizing the strengths of our work, including: (1) addressing an **underexplored yet important problem**; (2) providing **justified theoretical analysis**; and (3) demonstrating **practical relevance through experimental evidence**. We hope the following responses adequately address your concerns and further clarify the contributions of our work.
>
> ---
>
> **[W1]** We thank the reviewer for the thoughtful feedback. We agree that providing stronger evidence that instruction-tuned models (i.e., base models) already possess a “ignorance awareness” capability on natural data would further support the motivation of our work.
>
> With regards to base model’s IA capability, we have added a new analysis in Appendix C.1, where we evaluate the refusal behavior of base models on a set of real-world factual queries (the RWD dataset) concerning entities and events that occurred after the model’s knowledge cutoff date. These examples span public figures, geopolitical developments, latest news, etc. after the knowledge cut-off dates of both base models, which were not available during pretraining. Our results show that **base models under investigation exhibit strong calibration in their refusal behavior**, consistently responding with appropriate uncertainty or abstention.
>
> With regards to the applicability of SEAT on real-world dataset, we have already conducted experiments on the real-world dataset (RWD) in the original submission as shown in Table 2 that our method SEAT can not only be applied to the synthetic dataset, but also be applied to the real world dataset.
>
> ---
>
> **[W2]** We thank the reviewer for acknowledging that, in realistic deployment settings, particularly across industry use cases, practitioners typically have access only to their own task-specific fine-tuning data, not the original proprietary datasets used for alignment by model developers.
>
> If this assumption is relaxed (i.e., the user has access to the original alignment corpus and procedures), they could in principle either apply SEAT as we show or perform a full re-alignment after traditional fine-tuning. However, as discussed in the Introduction (new line 100), such re-alignment is often impractical in downstream scenarios due to limitations in data access and computational resources.
>
> Our goal is to demonstrate that SEAT remains robust and effective even under the realistic constraint of having only task-specific data. Access to additional alignment data would not conflict with SEAT’s effectiveness and robustness.
>
> ---
>
> **[W3]** Regarding the concern that the masking operation may impede learning, prior work (including the well-established lottery ticket hypothesis and many follow-up studies [1, 2]) has shown that not all parameters are necessary for effective learning, and that moderate sparsity does not hinder performance. Empirically, our FT scores remain near 1.0 across all settings (Table 2), indicating that SEAT does not impair learning on the target task.
>
> To further clarify the connection to IA preservation, we have added new analyses in Figure 3(a) and (b), along with accompanying discussion in the Ablation Study section. Specifically, we measure both the absolute activation drift and its projection along the IA direction for fine-tuning and unseen datasets. The results show that **sparsity does not impede learning**: **activations of fine-tuning data exhibit meaningful shifts**, confirming that the remaining trainable parameters are sufficient to adapt. In contrast, **change of activations for unseen data remain close to zero, demonstrating strong retention of the model’s original IA capability**.
>
> [1] The Lottery Ticket Hypothesis: Finding Sparse, Trainable Neural Networks, ICLR 2019
>
> [2] Sparse is Enough in Fine-tuning Pre-trained Models, ICML 2024

---

> ### Author Response · Authors · 2025-11-28
>
> **[W4]** As suggested by the reviewer, we have expanded our benchmark to LoRA variant (CLoRA (Lu et al., 2025)), EWC, and re-alignment methods: R-tuning (Zhang et al., 2024). Additionally, we added experience replay as a representative replay method. Full results are incorporated in the revised Table 2.
>
> When applying CLoRA, EWC or R-tuning, the model almost completely loses IA capability, with IDK rates on the unverifiable-knowledge dataset dropping from ~1.0 (base model) to less than 0.1 in most circumstances. Experience replay shows greater robustness in preserving IA, its performance remains significantly below SEAT, unstable, and highly dependent on the base model and fine-tuning dataset.
>
> Regarding DPO as a fine-tuning method, we conducted experiments on Llama-8B-Instruct, using PISTOL as the training set and either TOFU or RWD (the same real-world dataset referenced in W1) as negative IDK examples. The results show substantial performance degradation: the FT score drops from 1.0 to ~0.4, and the $\text{IDK}_{\text{HA}}$ score decreases from ~1.0 to ~0.65. This indicates both ineffective task learning and insufficient IA retention under standard DPO.
>
> These results strengthen our core message: preserving alignment-level meta-cognitive behaviours requires a method fundamentally different from traditional CL strategies, and SEAT explicitly targets this under-explored yet practically crucial capability. We have updated the experimental section to better emphasize this distinction and added the new baselines for completeness and clarity.
>
> We hope this expanded comparison adequately addresses the reviewer’s concerns and demonstrates that SEAT is not merely competitive but uniquely effective for the targeted problem setting.

---

### Official Review · Reviewer_WJrx · 2025-11-01

**Soundness:** 2
**Presentation:** 2
**Contribution:** 1
**Rating:** 2
**Confidence:** 3

**Summary:**

The paper proposes SEAT, a LLM fine-tuning method designed to preserve the model’s *Ignorance Awareness*, its ability to recognize and express uncertainty about unknown information, while learning new knowledge. The authors argue that conventional fine-tuning often causes activation displacement, which erodes this property and leads to hallucinations. SEAT addresses this issue with two key components: (1) *sparse tuning*, which limits activation drift during fine-tuning, and (2) *entity perturbation*, which mitigates entanglement by preventing inadvertent generalization to neighboring data points. Experiments on both real and synthetic datasets show that SEAT maintains ignorance awareness effectively, without sacrificing the ability to learn new information.

**Strengths:**

- [S1] **Timely and relevant topic.** Preserving uncertainty calibration and avoiding hallucinations with a focus on ignorance awareness during fine-tuning is an important challenge for LLM safety and reliability.
- [S2] **Combination of practical and theoretical perspectives.** The inclusion of both theoretical analysis (Section 3) and empirical experiments is a positive aspect, showing the authors’ effort to ground their approach in both intuition and formal reasoning.

**Weaknesses:**

- [W1] **Limited novelty in both problem formulation and methodology.** The idea of maintaining ignorance awareness during fine-tuning appears to overlap heavily with existing research in continual learning (as suggested in the Related Works section). Responding with “I don’t know” can be viewed simply as learning another data distribution, so it is unclear if this warrants a distinct formulation. As a result, the proposed techniques, sparse tuning and entity perturbation, appear to be adaptations of existing methods: the former has been explored in continual learning (e.g., [A]) and the latter parallels adversarial perturbation and decision-boundary preservation methods from machine unlearning (e.g., [B]). Therefore, the methodological contribution feels incremental.
- [W2] **Inadequate experimental comparisons.** The experimental section claims that directly comparable baselines are unavailable, yet it seems existing continual fine-tuning methods can be tested seamlessly within the framework. Comparing SEAT only to Full FT and Sparse FT baselines does not convincingly demonstrate its advantages.
- [W3] **Weak and unclear theoretical development.** Theoretical results in Section 3 (Theorems 1–3) are difficult to follow and contribute limited new insight. The early theorems state rather trivial results about sparse updates, while the main theorem (Theorem 3) relies on an assumption of Lipschitz continuity for the score functional, a property that does not hold for self-attention networks [C]. Moreover, the argument appears to depend on an empirical observation from unrelated unlearning work [D], weakening its rigor. The section would benefit from a clearer statement of assumptions and a more intuitive discussion of implications.

[A] Continual Learning with Node-Importance based Adaptive Group Sparse Regularization. NeurIPS 2020.\
[B] Learning to Unlearn: Instance-wise Unlearning for Pre-trained Classifiers. AAAI 2024.\
[C] The Lipschitz Constant of Self-Attention. ICML 2021.\
[D] LLM Unlearning via Neural Activation Redirection. NeurIPS 2025.

**Questions:**

- [Q1] In the ablation studies, it appears that the configuration “Full FT + KL w/o EP” (i.e., without both sparse tuning and entity perturbation) is missing. Would including this help clarify the relative contribution of each component?
- [Q2] Citation formatting seems inconsistent (mixing \cite{} and \citep{}). Please ensure consistent use of citation style throughout the paper.

---

> ### Author Response · Authors · 2025-11-27
>
> We thank the reviewer for their thoughtful feedback and for recognizing that our work provides a **theoretically grounded approach** to a **timely and important problem**. We hope the following responses address your concerns.
>
> ---
>
> **[W1]** We appreciate the opportunity to clarify both the conceptual and methodological novelty of our work.
>
> **First**, we reckon that **retaining original IA alignment (as a broad meta-cognition capability) is a fundamentally different problem to “learning another distribution” of “I don’t know” (IDK) responses**. IA is a general capability to recognize the limits of LLM’s knowledge (IDK can be a suitable response to any task distribution) and should generalize beyond the training distribution to unbounded open-ended or out-of-distribution inputs that it has not seen before. Existing continual learning assumes well-defined task boundaries and data manifolds, whereas IA preservation requires maintaining **self-calibration as a broad capability across an unbounded open-ended input space**. This fundamental difference underlies the failure of existing continual learning methods to preserve IA.
>
> **Second**: Conceptual, theoretical and methodological difference from [A]:
>
> - **Different objective**: [A] mitigates forgetting of bounded task-specific data (with discrete labeled tasks and therefore a clear task boundary) while SEAT preserves a broad capability (questions that the LLM is unable to answer is unbounded).
>
> - **Different mechanism**: because of its task-specific nature, [A] tracks parameter importance per task and reinitializes unimportant nodes. In contrast, SEAT preserves IA without any task partitioning, node tracking, or reinitialization.
>
> - **Different theoretical foundation**: [A] is grounded in stability–plasticity optimization for small/medium networks. SEAT is derived from the linear representation hypothesis of modern LLMs, constraining over-drift of its residual stream activations to preserve key abilities.
>
> - SEAT’s sparse tuning masks are **data-agnostic and unstructured** (we show that even random masks suffice) highlighting that SEAT operates on representational stability rather than task-specific weight freezing, a qualitatively distinct principle from [A].
>
> **Third**: Distinct motivation, method and implementation from [B]:
>
> We agree that using perturbed inputs to anchor model behaviour, originally proposed in [2], is a general principle that has inspired many works including [B] and SEAT. However, SEAT features significantly different motivation, implementation and implications compared to [B].
>
> - **Different goal**: [B] generates adversarial examples of forget data to mitigate forgetting on retained data (two different data splits). SEAT perturbs entity name in the knowledge to be acquired to prevent semantic spillover.
>
> - **Different mechanism**: [B] employs iterative Projected Gradient Descent to create adversarial inputs (computationally heavy iterative process and requires 20–200 adversarial examples per input). SEAT needs only one randomly generated entity name, requiring no gradient-based search and negligible additional cost.
>
> - **Different objective**: [B] minimizes cross-entropy on adversarial samples. SEAT minimizes KL divergence to prevent semantic spillover of the fine-tuning data , a calibration objective absent in [B]’s formulations.
>
> In short, SEAT introduces a **new capability-preserving fine-tuning paradigm**, distinct from prior CL works for task or knowledge retention. Its novelty lies in formalizing and preserving IA as an epistemic calibration capability, and in proposing a theoretically grounded activation-anchoring and entity-aware mechanism to operationalize this goal within modern LLM architectures.
>
> [1] Do Large Language Models Know What They Don't Know? ACL 2023
>
> [2] Adversarial Examples Are Not Bugs, They Are Features. NeurIPS 2019

---

> > ### Author Response · Authors · 2025-11-27
> >
> > **[W2]** We thank the reviewer for this comment and for highlighting the importance of strong baselines. We did not compare with existing continual learning methods because SEAT targets preservation of a meta-cognitive capability (IA) rather than task-specific knowledge. That said, we agree that, to fully understand the value of SEAT, it is helpful to examine whether existing continual learning could serve as viable alternatives for preserving alignment as well.
> >
> > To directly address the reviewer’s concern, we have expanded our benchmark to include **three additional categories** of new baselines: (1) representative continual learning approaches: CLoRA (Lu et al., 2025) and EWC (Kirkpatrick et al., 2017; Loke et al., 2025); (2) re-alignment methods: R-tuning (Zhang et al., 2024); and (3) experience replay. We report the full results in the revised Table 2.
> >
> > The findings substantiate our claim: When applying CLoRA, EWC or R-tuning, the model **almost completely loses IA capability**, with IDK rates on the unverifiable-knowledge dataset dropping from ~1.0 (base model) to less than 0.1 in most circumstances. Experience replay shows greater robustness in preserving IA, however, its performance remains **significantly below SEAT**, **unstable**, and **highly dependent on the base model and fine-tuning dataset**. This suggests that while traditional CL methods are effective for task-level knowledge retention, they lack robustness in preserving original IA alignment.
> >
> > These results strengthen our core message: preserving alignment-level meta-cognitive capabilities requires a method fundamentally different from traditional CL strategies, and SEAT explicitly targets this under-explored yet practically crucial capability. We have updated the experimental section to better emphasize this distinction and added the new baselines for completeness and clarity.
> >
> > We hope this expanded comparison adequately addresses the reviewer’s concerns and demonstrates that SEAT is not merely competitive but uniquely effective for the targeted problem setting.

---

> ### Author Response · Authors · 2025-11-28
>
> **[W3]**
>
> - **Clarifying the Lipschitz assumption in original Theorem 3**
>
> The reviewer notes that original Theorem 3 *“relies on an assumption of Lipschitz continuity for the score functional, a property that does not hold for self-attention networks [C].”* First, the score functional has no self-attention component. Second, we believe this comment confuses two unrelated notions of Lipschitzness.
>
> Original Theorem 3 does not assume that the Transformer or self-attention block is globally input-Lipschitz over unbounded domains, the core claim of [C]. Instead, it assumes: (1) A proper scoring rule is Lipschitz in probability on a bounded interval $[\delta, 1-\delta]$: a standard property we proved in the original Lemma 5; (2) The ignorance read-out head $f_\theta(h)$ is Lipschitz with respect to the residual representation $h$ (already proved in Corollary 4 and elaborated in the proof of Theorem 3) which is a local assumption, not on the underlying Transformer. This assumption is representation-level and do not require global Lipschitz continuity over the entire input domain.
>
> Hence, **the assumptions in Theorem 3 and the findings of [C] are conceptually orthogonal and do not conflict**.
>
> - **Clarifying the role of original Theorems 1 and 2**:
>
> We agree with the reviewer that original Theorems 1 and 2 are conceptually straightforward. This simplicity is intentional as we want them to lay the theoretical foundation for original Theorem 3 (i.e., sparsity constrains activation drift and tightens the bound given by $\epsilon$ in Theorem 3).
>
> The novelty we claim is the IA formalization and the end-to-end IA preservation under SEAT, not the algebra of masking itself. As explained from the original line 224 (i.e., new line 245), we extended the beneficial role of sparsity and show that it anchors the original alignment via mitigating over-drift due to fine-tuning, a result corroborated empirically in Figure 1(d).
>
> To address the reviewer’s concern, we have reorganized the presentation of our theoretical results. Specifically, we now promote the original Theorem 3 to Theorem 1 to emphasize its central role in the paper. The original Theorems 1 and 2 have been re-labeled as Lemma 3 and Lemma 6, respectively, and moved to the Appendix, where they support the proof of the new Theorem 1.
>
> We believe this restructuring better highlights the theoretical foundation of our work and hope it adequately addresses the reviewer’s feedback.
>
>
> - **Clarifying the role of empirical observation [D]**
>
> The reviewer noted that the *"argument appears to depend on an empirical observation from unrelated unlearning work [D].”*
>
> To clarify, [D] is the motivation of this work: it highlights the empirical fragility of IA alignment under standard fine-tuning, even with small datasets. The theoretical results themselves are not related to or dependent on [D].
>
> Our analysis provides a formal explanation for this phenomenon: unconstrained updates can produce large residual drift ($\varepsilon$), leading to large IA degradation in new Theorem 1. We also added Appendix C.1, which: (1) empirically demonstrates the existence of IA in base models, and (2) shows its collapse under conventional fine-tuning (see Table 2), reinforcing that the phenomenon motivating our theory is real and measurable.
>
> We would be grateful if the reviewer could clarify what specific aspect seems to “weaken rigor,” and we are happy to refine further.
>
> ---
>
> **[Q1]** Without entity perturbation, the KL loses point (i.e., the second loss term in loss function disappears). The result is then the standard Full FT in Table 2. We restructured Table 3(b) to make this point clearer in the updated version.
>
> **[Q2]** We thank the reviewer for your close attention. We have updated it in the revised version.

---

### Author Response · Authors · 2025-11-28

**Dear Reviewers, ACs, and PCs,**

We sincerely thank you for your time, thoughtful feedback, and constructive suggestions. We are encouraged by the reviewers’ recognition that our work: (1) addresses an **important and under-explored problem** (WJrx01, ytXg, ZnQP); (2) presents a **theoretically grounded approach** (WJrx01, ytXg, ZnQP), and (3) demonstrates **practical relevance through experimental evidence** (ytXg, ZnQP).

We have made substantial revisions to the manuscript, including **expanded baselines**, **additional ablation studies**, **streamlined theory section**, and **enhanced qualitative examples and discussions**, to address **all reviewers’ concerns** and **improve clarity** throughout.

---

### **Summary of Revisions and New Results**

**Problem Setting Clarification**:
To address concerns regarding the problem formulation, particularly (1) the distinction between preserving original ignorance awareness (IA) as a meta-cognitive alignment capability versus retaining task-specific knowledge (WJrx01), and (2) the relation to re-alignment techniques (8Sdd), we:
- Revised the Introduction to improve clarity over problem setting and motivation.
- Expanded the baseline comparison with three new categories (continual learning-style methods, light re-alignment, and replay technique). Results support the superior performance of SEAT, reinforcing that this is a distinct and novel problem.

**Theory Presentation Streamlining**:
We significantly revised the theoretical analysis section to improve readability and clarity (WJrx01, 8Sdd). Specifically, we:
- Promoted the original Theorem 3 to Theorem 1 to foreground its central importance.
- Re-labeled the original Theorems 1 and 2 as Lemmas 3 and 6 and moved them to the Appendix to clarify their supporting role in the core proof and improve overall presentation clarity.
- Addressed confusions regarding key assumptions and theoretical implications in the accompanying discussion below.

**Sparse Training and Learning Effectiveness**:
In response to concerns about whether sparsity may impede learning (ytXg, 8Sdd), we added new analysis and results (Figure 3(a), (b), Ablation Study) demonstrating that:
- The model learns meaningfully from the fine-tuning data with significant activation updates.
- Activation drift for unseen data remain close to 0, indicating strong retention of IA.
- Highlight the clear separation between seen and unseen data along the IA direction in activation space (Figures 2 and 4–8), to better illustrate the strong alignment-preserving benefits of SEAT.

**IA Calibration of Real-World Data**:
To address the request for evaluating IA on real-world data (ytXg), we included new analysis showing that base models exhibit strong ignorance calibration on real-world examples, strengthening the empirical grounding of our problem setup.

**Improved Presentation**:
To enhance overall clarity, we:
- Added Figure 1 to visually convey the core mechanism of SEAT.
- Included additional qualitative examples to illustrate the preservation of IA post fine-tuning.

---

### **Request for Further Consideration**

We believe your feedback has strongly improved our paper. All additions are highlighted in **blue**. We remain grateful for your detailed engagement and kindly ask for your further feedback and inquiry. We also provide **detailed responses to shared reviewer concerns in the comment below**.

— The Authors

---

### Meta-Review · Area_Chair_g8qs · 2025-12-23

**Summary:**

This paper proposes an LLM fine-tuning method to deal with catastrophic forgetting by preserving the model’s ignorance awareness.

Strengths:
(1) addressing an important and under-explored problem; (2) a theoretically grounded approach

Weaknesses:
(1) limited technical novelty; (2) inadequate experimental evaluation including missing strong baselines; (3) lack of justification of the underlying assumption regarding the pretrained model; and (4) applicability and practicality of the proposed method.

**Reviewer Concerns:**

The authors addressed some concerns and weaknesses, including empirical evaluations and underlying assumptions.

**Reviewer Scores:**

It is unlikely that reviewers will change their scores.

---

### Decision · Program_Chairs · 2026-01-26

Reject